# A Case of ^123^I-Metaiodobenzylguanidine Scintigraphy-Negative Pheochromocytoma with a Tumor-Developing Mutation in the *RET* Gene

**DOI:** 10.3390/jcm11154624

**Published:** 2022-08-08

**Authors:** Haremaru Kubo, Yuya Tsurutani, Takashi Sunouchi, Yoshitomo Hoshino, Rei Hirose, Sho Katsuragawa, Noriko Kimura, Jun Saito, Tetsuo Nishikawa

**Affiliations:** 1Endocrinology and Diabetes Center, Yokohama Rosai Hospital, Yokohama 222-0036, Japan; 2Division of Clinical Research, Pathology Section, National Hospital Organization, Hakodate Hospital, Hakodate 041-8512, Japan

**Keywords:** pheochromocytoma, GAPP score, *RET* gene, MIBG scintigraphy

## Abstract

Pheochromocytoma (PCC) is rare catecholamine-producing endocrine tumor that metastasizes in approximately 10% of cases. As a functional imaging of PCC, ^123^I-metaiodobenzylguanidine (MIBG) scintigraphy was established, and some cases of PCC exhibit negative accumulation on MIBG scintigraphy, indicating a high risk of metastasis. Additionally, germline genetic variants of PCC are evident in approximately 30% of cases, although the genotype-phenotype correlation in PCC, especially the association between genetic mutations and MIBG scintigraphy, remains unclear. A 33-year-old man was admitted to our hospital for further examination for hypertension. He was diagnosed with sporadic PCC, and left adrenalectomy was performed. The adrenal tumor was negative on MIBG scintigraphy. Histology of the tumor revealed a moderately differentiated PCC. Target gene testing revealed a mutation in *RET* (c.2071G > A). This mutation has been reported to be a tumor-developing gene involved in the pathogenesis of PCC. Moreover, the *RET* mutation is the only gene mutation reported in a previous study of PCC with negative results on MIBG scintigraphy, except for the *SDHB* gene mutation, which is a common mutation in metastatic PCC. Correctively, the present *RET* gene mutation may be associated to MIBG-scintigraphy negative PCC and its pathophysiology. Clinicians should follow such cases more cautiously in clinical practice.

## 1. Introduction

Pheochromocytoma (PCC) and paraganglioma are rare catecholamine-producing neoplasms arising from the adrenal medulla and sympathetic/parasympathetic paraganglia. They become metastatic in approximately 10–15% of cases, and life-long follow up is needed [1,2]. Although PCC mainly occurs sporadically, approximately 30–40% of cases are reported to be accompanied by germline mutations [1,3]. To date, approximately 20 susceptibility genes for PCC have been reported, including *RET, VHL, MAX, TMEM127, SDHB, CSDE1*, and *HIF2A* [1,3,4]. In particular, *SDHB* mutations are associated with a high metastatic frequency [1] and poor differentiation [5]. In contrast, *RET* mutations are mainly recognized in patients with multiple endocrine neoplasia type 2 (MEN2), which is a hereditary condition associated with three primary types of tumors: medullary thyroid tumors, parathyroid tumors, and PCC. They typically exhibit well-differentiated (i.e., low risk of metastasis) tumors of the adrenal glands [6].

^123^I-metaiodobenzylguanidine (MIBG) scintigraphy has been established as the gold standard method for the functional imaging of PCC [7,8]. MIBG is a guanidine analog, like norepinephrine, that enters chromaffin cells by active uptake via transporters, and it is stored in the neurosecretory granules through the vesicular monoamine transporter (VMAT) [7,8]. Through this mechanism, at least 56% and 70%, up to 83–100% and 95–100% in sensitivity and specificity, PCC are positive on MIBG scintigraphy [9], although PCC negative on MIBG scintigraphy reportedly exhibits a higher metastatic rate [7,8].

Herein, we report the case of a 33-year-old man diagnosed with PCC. His tumor demonstrated negative accumulation on MIBG scintigraphy and an intermediate level of differentiation on pathological examination. Additional gene testing revealed a single-nucleotide polymorphism (SNP) in *RET*, which is reported to be associated with tumor development in PCC. This case provides new insights into the association between *RET* mutations and MIBG scintigraphy.

## 2. Case Presentation

A 33-year-old Japanese man was referred to our hospital for further evaluation of hypertension. He started experiencing headache-related nausea when he was 30 years old, and this was accompanied by palpitations starting when he was 32 years old. He exhibited severe hypertension (systolic blood pressure as high as 200 mmHg), and medication was started while he was examined at a clinic. He was also diagnosed with hypertensive retinopathy prior to admission.

Computed tomography (CT) revealed a 25 mm-diameter tumor in the left adrenal gland (Figure 1) accompanied by a double left renal vein. Biochemical tests (Table 1) revealed high levels of plasma renin activity (2.2 ng/mL/h), plasma aldosterone (186 pg/mL), and serum and urinary catecholamine (serum noradrenaline = 5.8 ng/mL, dopamine = 0.14 ng/mL, urinary adrenaline = 54.8 µg/day, urinary noradrenaline = 3650 µg/day, urinary dopamine = 1200 µg/day, and urinary and normetanephrine = 4.0 mg/day). His hemoglobin A1c (HbA1c) was 6.0%, and a 75-g oral glucose tolerance test revealed an impaired glucose tolerance [116(0)–268(30)–139(120) mg/dL(min)]. Additionally, a 1-mg dexamethasone suppression test was negative (cortisol = 1.7 mg/dL). These data were suggestive of PCC, although MIBG scintigraphy did not demonstrate any radioactivity in the left adrenal tumor (Figure 2). Magnetic resonance imaging (MRI) demonstrated left adrenal isointense activity on T2-weighted images, which is consistent with PCC (Figure 3A). Additionally, ^18^F-fluoro-2-deoxy-D-glucose positron emission tomography/CT demonstrated a standardized uptake value of 5.6 in the adrenal tumor (Figure 3B), and there was no evidence of metastasis. Based on these results, the patient was diagnosed with PCC of the left adrenal gland. We initiated alpha-blockade therapy with doxazosin and adjusted the medication of antihypertensive drugs preoperatively: up to 24 mg/day doxazosin and 2.5 mg/day bisoprolol. Left adrenalectomy was performed based on the clinical diagnosis of MIBG scintigraphy-negative PCC. The patient became medication-free immediately after the surgery, and he achieved good control of his blood pressure.

The left PCC adrenal tumor exhibited a brownish, gel-like, soft-appearance, and it was 50 × 28 × 15 mm^3^ in size. Microscopically, the tumor cells displayed nuclear hyperchromasia with scant cytoplasm and a perivascular pseudorosette pattern, suggesting a potentially high risk of malignancy (Figure 4). SDHB immunostaining of the tumor was positive, suggesting that the tumor did not have a mutation in the *SDHB* gene. The GAPP score [4] was 4 points (pseudorosette pattern: 1 point; high-cellularity [230 cells/unit]: 1 point; and Ki67 labeling index [7%]: 2 points) as category of moderately differentiation. No vascular or lymphatic invasions were observed. This moderately differentiated GAPP score (3–6 points) is reported to be significantly associated with disease recurrence (Hazad Ratio: 3.367) compared with well-differentiated score (1–2 points) [10], indicating an increased level of future risk of metastasis or recurrence. We continued careful follow-up for 3 years after the operation, and this resulted in no metastasis or recurrence.

After informed consent was obtained from the patient, we conducted targeted germline gene testing covering three molecular types of pheochromocytoma [11]; (1) the pseudohypoxia group (*SDHB*, *VHL* and *HIF2A*), (2) the Wnt signaling group (including *CSDE1*) and (3) the kinase signaling group (including *RET, MAX* and *TMEM127*) for detecting germline mutation of PCC with the methods shown in Material and Method section. Target gene testing revealed an SNP in the *RET* gene (c.2071G > A, p.Gly691Ser). Previous studies report that this mutation is related to the occurrence of PCC in patients diagnosed with MEN2A [10,11]. However, there was no familial history of MEN2-associated diseases in the present case, including the absence of other PCC, thyroid nodules, and parathyroid dysfunction. This mutation was previously reported to exhibit a functional change in the phosphorylation pattern, which may enhance signal transduction in in silico analysis [12]. However, this was reported as a “conflicting interpretation” in Clinvar (NM_020975.6). Therefore, we could not diagnose the patient with MEN2. To the best of our knowledge, this is the first report of a case of sporadic PCC with the present *RET* mutation.

## 3. Materials and Methods

Genomic DNA was extracted from whole-blood samples using NucleoSpin^®^ Blood L (Takara Bio, Shiga, Japan). The coding exons of *MAX*, *TMEM127*, *SDHB*, *CSDE1*, and *HIF2A* genes were amplified according to the Tks Gflex DNA Polymerase protocol (Takara Bio, Shiga, Japan). The PCR conditions were as follows: 40 cycles of 10 s at 98 °C, 15 s at 60 °C, and 1 min at 68 °C. Primers were designed as Appendix A. Bands of the PCR products were cut from agarose gel and purified using Wizard PCR clean-up system (Promega, Madison, WI, USA). These methods were conducted by a well-experienced technician. Thereafter, these sequencing analyses was outsourced to Eurofins Genomics (Tokyo, Japan).

*RET* and *VHL* genes sequences were outsourced commercially from the step of extracting gDNA from whole blood sample through LSI Medience Corporation (Tokyo, Japan).

## 4. Discussion

Herein, we describe a case of MIBG scintigraphy-negative and intermediate-differentiated PCC with an SNP in the *RET* gene (c.2071G > A, p.Gly691Ser).

PCC exhibits an abundant expression of catecholamines and related vesicle transporter systems. This phenomenon can be visualized using MIBG scintigraphy. Therefore, MIBG scintigraphy is routinely used as the gold-standard diagnostic tool for PCC [13]. MIBG scintigraphy has a sensitivity as high as 83–90% and a specificity as high as 95–100% for PCC [9], but some of the cases have negative results. In particular, the decreased expression of VMAT is thought to be the main cause of negative accumulation on MIBG scintigraphy due to decreased amins uptake. This association is already described in neuroendocrine tumors [14], though it has not been proven yet in PCCs [15] probably due to small-sized samples and disease rarity. However, comparing three major types of PCCs provided by the Cancer Genome Atlas (TCGA) project [11], the kinase signaling group including *RET* mutation shows decreased VMAT expression than other groups [16,17]. Otherwise, in previous reports, the prevalence of metastatic PCC tends to be increased by up to 50% in MIBG scintigraphy-negative cases [18,19]. These reports suggest that poor differentiation or an increased risk of metastasis may be associated with MIBG scintigraphy-negative PCCs. Additionally, MIBG scintigraphy-negative PCC exhibits a higher frequency of the *SDHB* mutation, suggesting aggressive disease behavior, than MIBG scintigraphy-positive PCC [15,19,20,21]. To support this, it is suggested that the differences in VMAT expression may signify a difference in cellular dedifferentiation and tumor aggressiveness [11]. Furthermore, in a previous report of gene testing conducted on PCC [18], the *RET* mutation was observed only in MIBG scintigraphy-negative PCC, except for the *SDHB* mutation. Taking into consideration the clinical data and these previous reports, the combination of possible decreased VMAT expression associated to *RET* mutation and pathological moderately immature differentiation might contribute to the MIBG-negativity in the present case. In short, it is suggested that the combination of *RET* gene mutations and immature differentiation of the tumor may be associated with MIBG scintigraphy-negative PCC when it does not also have express the *SDHB* mutation. We should conduct an even more careful follow-up of patients with PCC exhibiting negative accumulation on MIBG scintigraphy.

As to the *RET* mutation in this case, this SNP is classified as “conflicting interpretations of pathogenicity” in Clinvar, and its function remains unclear. *RET* mutations are typically associated with MEN2A mutations. In patients with MEN2A, PCC is reported to have a low metastatic rate (<5%) and show relatively well-differentiation [22,23], although there are some case reports of metastatic PCC with *RET* gene mutations [22,24]. Furthermore, the *RET* mutation in the present case is also associated with the metastasis of medullary thyroid carcinoma [25,26], indicating the potential for tumor development and malignancy. Moreover, this mutation tends to be observed in PCC in patients diagnosed with MEN2A [12,27]. The mutation observed in the present case is located in the tyrosine kinase domain, which regulates cell proliferation [28], and it has been shown to exhibit predicted changes in the phosphorylation pattern by in silico analysis [29,30]. Therefore, it is probable that the present *RET* mutation has some effects on the kinase signaling in some way. In summary, these recent reports indicate that the *RET* gene mutation in the present case exhibits tumor-developing potential, especially PCC-developing potential, via the regulation of cell proliferation. As a limitation, there is a possibility that this mutation exists accidentally in the present case. Therefore, the association between the present *RET* mutation and the phenotypes in the present case is one of the hypotheses. Further analysis about the genotype-phenotype correlation is required.

Pathologically, the tumor in the present case exhibited a pseudorosette formation, which rarely occurs in PCC. A pseudorosette pattern usually suggests immature tumors, which are mainly observed in pseudopapillary tumors of the pancreas and in some pituitary tumors [2,31]. Previously, a case of poorly differentiated colorectal paraganglioma exhibiting a pseudorosette pattern and demonstrating a negative result on SDHB immunostaining, suggesting a *SDHB* gene mutation, was reported [32]. Additionally, a case of intermediately differentiated PCC that was positive on SDHB immunostaining exhibited this pseudorosette pattern [33], suggesting the absence of a mutation in the *SDHB* gene, although target germline gene testing was not conducted. Taking into consideration our case and these reports, PCC or paraganglioma exhibiting a pseudorosette pattern may have a germline mutation, such as a mutation in *SDHB* or a mutation in *RET*, accompanied by relatively poor differentiation. However, only a few cases have been reported, and the association between pathological findings and genotype remains unclear and requires further investigation. Additionally, some other genes which we could not confirm sequence might have effect on the pathological phenotypes.

Finally, the *RET* mutation in the present case has also been reported in patients with vesicoureteral reflux from a urological anatomical variant [34]. In anatomical studies, the mechanism of variations in the renal anatomy, such as the renal vein, renal artery, and ureter, are intimately involved and often occur in the same individual [35,36]. Based on these reports, the double renal vein observed in the present case also suggests that the tumor expresses the phenotype of this *RET* mutation.

## 5. Conclusions

We report a case of MIBG scintigraphy-negative PCC accompanied a *RET* mutation. A *RET* mutation in the present case exhibits tumor-developing potential and may be associated with MIBG scintigraphy-negative PCC. MIBG scintigraphy-negative PCC has an increased risk of metastasis and more careful follow-up is needed in such cases. The genotype-phenotype correlation in PCC remains to be elucidated.

## Figures and Tables

**Figure 1 jcm-11-04624-f001:**
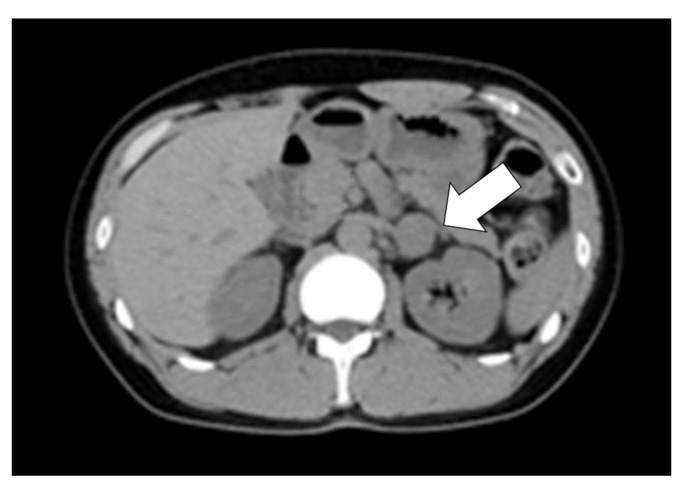
Abdominal computed tomography (CT) image of the patient in this case. Abdominal CT image during the first outpatient visit. Left adrenal adenoma (25 mm in diameter and 40 household-units) is shown (white arrow).

**Figure 2 jcm-11-04624-f002:**
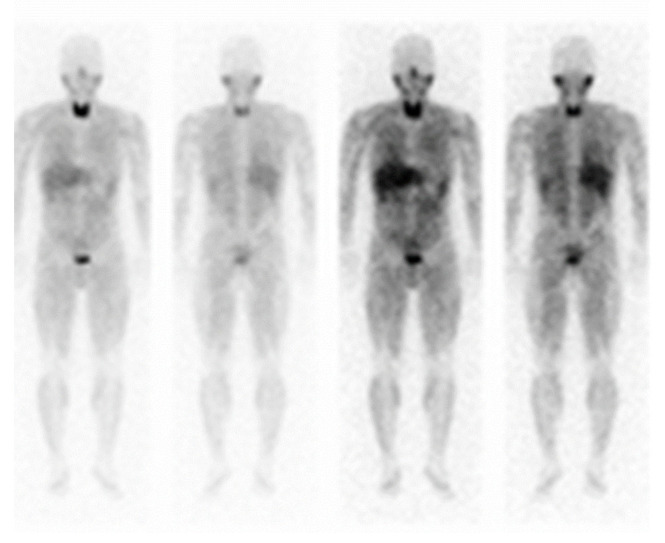
^123^I-metaiodobenzylguanidine (MIBG) scintigraphy of the patient from the present case. Functional imaging of the patient with MIBG-negative pheochromocytoma. The left adrenal tumor was not detected.

**Figure 3 jcm-11-04624-f003:**
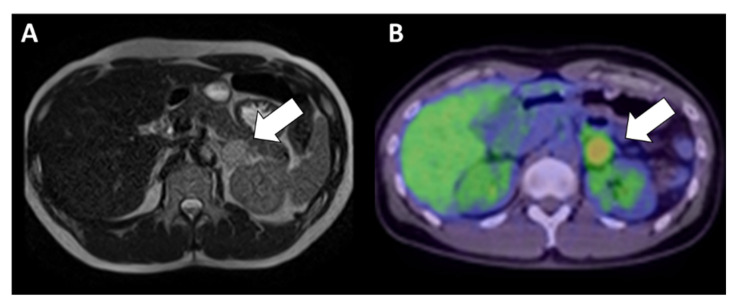
Abdominal images of the patient in the present case. Abdominal magnetic resonance imaging (MRI) results of the adrenal mass are shown as a T2-weighted image (**A**). The tumor exhibited a moderate intensity. Fluorodeoxyglucose positron emission tomography imaging (**B**) revealed slight accumulation (maximum standardized uptake value = 5.6) of the adrenal mass.

**Figure 4 jcm-11-04624-f004:**
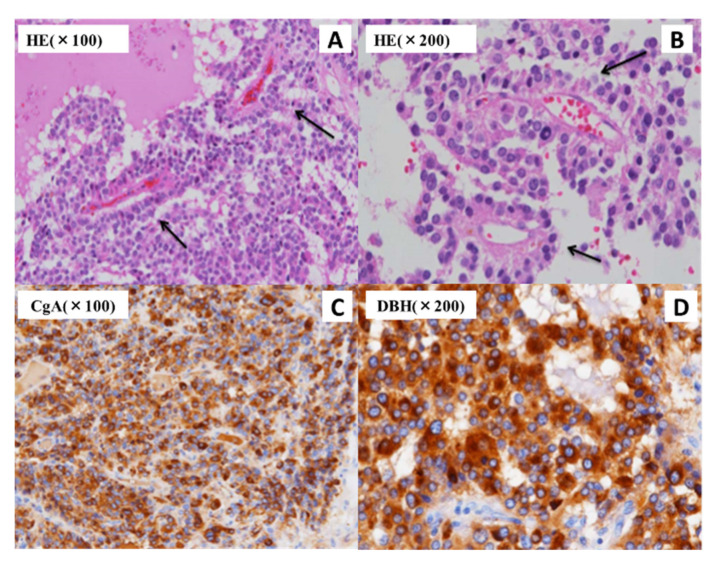
The pathological findings in the present case reveal an adrenal tumor. Small-sized cell tumors with large round nuclei, scant cytoplasm, and high-intensity chromatin are observed using hematoxylin-eosin (HE) staining. A perivascular pseudorosette pattern (black arrow) is also observed (**A**,**B**). The immunohistochemical detection of chromogranin A (CgA) (**C**) and dopamine β-hydroxylase (DBH) (**D**) demonstrated positive results in the histopathological analysis.

**Table 1 jcm-11-04624-t001:** Laboratory data on admission.

Parameter	Value	Unit	Reference Range	Parameter	Value	Unit	Reference Range
**<Blood chemistry data>**				**<Endocrinological data>**			
Total protein	6.6	g/dL	6.7–8.3	ACTH	30.1	pg/mL	7.2–63.3
Albumin	4.4	g/dL	4.1–5.1	Cortisol	10.7	mg/dL	1.0–19.3
AST	18	U/L	13–30	Growth hormone	0.06	ng/mL	0.13–9.88
ALT	20	U/L	10–42	IGF-1	132	ng/mL	103–287
γ-GTP	14	U/L	13–64	PRA	2.2	ng/nL/hr	0.3–2.9
BUN	8.9	mg/dL	8.0–20.0	PAC	18.6	ng/dL	3.0–15.9
Creatinine	0.77	mg/dL	0.65–1.07	LH	3.44	mIU/mL	2.20–8.40
Na	140	mEq/L	138–145	FSH	10.2	mIU/mL	1.8–12.0
K	4.2	mEq/L	3.6–4.8	Prolactin	7.24	ng/mL	1.50–9.70
Cl	106	mEq/L	101–108	TSH	1.32	μIU/mL	0.500–5.000
Ca	9	mEq/L	8.8–10.1	Free T3	2.6	pg/mL	2.3–4.3
IP	3.1	mEq/L	2.7–4.6	Free T4	1	ng/dL	0.9–1.7
Mg	2.1	mg/dL	1.8–2.6	Calcitonin	1.34	pg/mL	<5.15
CRP	<0.01	mg/dL	0.00–0.14	PTH-intact	6.12	pg/mL	10–65
BNP	10.2	pg/mL	0.0–18.4	Adrenaline	0.05	ng/mL	<0.10
Total cholesterol	185	mg/dL	130–219	Dopamine	0.14	ng/mL	<0.03
Triglyceride	49	mg/dL	36–150	**<Urinary data>**			
HDL-cholesterol	80	mg/dL	41–67	Adrenaline	54.8	μg/day	3.0–41.0
LDL-cholesterol	94	mg/dL	70–139	Noradrenaline	3650	μg/day	31.0–160.0
HbA1c	6.0	%	4.6–6.2	Metanephrine	0.2	mg/day	0.05–0.20
Blood glucose	127	mg/dL	10–110	Normetanephrine	4	mg/day	0.10–0.28
CPR	15.5	ng/mL	0.80–2.50	Cortisol	135.5	μg/day	4.3–176.0
Anti-GAD antibody	<0.5	U/mL	<5.0	Aldosterone	11.2	μg/day	1.0–19.3

Abbreviations: AST, aspartate aminotransferase; ALT, alanine aminotransferase; γ-GTP, γ-glutamyl transpeptidase; BUN, blood urine nitrogen; CRP, C-reactive protein; BNP, B-type natriuretic peptide; HDL, high-density lipoprotein; LDL, low-density lipoprotein; HbA1c, hemoglobin A1c; CPR, C-peptide immunoreactivity; GAD, glutamic acid decarboxylase; ACTH, adrenocorticotropic hormone; IGF-1, insulin-like growth factor-1; PRA, plasma renin activity; PAC, plasma aldosterone concentration; LH, luteinizing hormone; FSH, follicle-stimulating hormone; TSH, thyroid-stimulating hormone; FT3, free triiodothyronine; FT4, free thyroxine; PTH-intact, intact parathormone.

## Data Availability

Not applicable.

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
