# Peer review of "A Case of 123I-Metaiodobenzylguanidine Scintigraphy-Negative Pheochromocytoma with a Tumor-Developing Mutation in the RET Gene"

_jcm, 2022, doi:10.3390/jcm11154624_

Round 1
Reviewer 1 Report
Kubo et al., present an interesting and novel case report of a patient with pheochromocytoma carrying a RET mutation with negative MIBG functional imaging which can in some cases be an indicator of increased metastasis. This is an in-depth case report and the paper is particularly well-written, the figures and tables are clear and presented to a high standard.
Minor comment: Prolactin typo (please change prolactine => prolactin in Table 1
Author Response
Point-by-Point Responses to the Comments:
Dear Editors and Reviewers
Journal of Clinical Medicine
We wish to submit a revised our article for publication in Journal of Clinical Medicine, titled “A case of 123I-metaiodobenzylguanidine scintigraphy-negative pheochromocytoma with a tumor-developing mutation in the RET gene.” (Manuscript ID: jcm-1840781).
We have carefully considered the comments provided by the reviewers, and our point-by-point responses to each are included in response letter shown below. The respective changes made to the manuscript are written with “Track Changes” function as instructed with some minor corrections for language, grammar, and consistency. Revised manuscript is attached as Word file and please see the content. Thank you very much for reviewing our manuscript and offering valuable advice.
Reviewer #1:
Kubo et al., present an interesting and novel case report of a patient with pheochromocytoma carrying a RET mutation with negative MIBG functional imaging which can in some cases be an indicator of increased metastasis. This is an in-depth case report and the paper is particularly well-written, the figures and tables are clear and presented to a high standard.
Minor comment: Prolactin typo (please change prolactine => prolactin in Table 1
Reply:
We apologize for our mistyping and thank you very much for suggestion. We subsequently revised manuscript in the Table 1 (page 3).
We believe that these changes have greatly improved the quality of our paper, and we wish to thank you for your consideration. I look forward to hearing from you.
Sincerely,
Haremaru Kubo and Yuya Tsurutani
Endocrinology and Diabetes Center, Yokohama Rosai Hospital, Japan 3211 Kozukue-cho, Kouhoku-ku, Yokohama, Kanagawa, 222-0036, Japan.
Tel: +81-45-474-8111
Fax: +81-45-474-8323
e-mail: yuya97tsuru1055@gmail.com
Reviewer 2 Report
The authors report a case of an MIBG negative pheochromocytoma where germ-line mutation analysis revealed a well-known polymorphism of the RET protooncogene. The clinical relevance of the Gly691Ser polymorphism is debated and it cannot be regarded as a pathogenic RET mutation. It is definitely not regarded as a causative mutation for the MEN2 syndrome.
Based on the inconclusive results on Gly691Ser - in my view - its pathogenic relevance cannot be established in this case either. The link between Gly691Ser and pheochromocytoma in this case is merely hypothetic and could also be an accidental association. It is unclear how the MIBG negativitiy of the PCC could be related to this polymorphism. The authors report no signs of MEN2.
The presentation of the methods is inadequate.
Minor points:
- the normal ranges of hormones should be given
- the authors state in the introduction that 70 % of PCC are MIBG positive, but later in the discussion they mention 90-100 % sensitivity and specificity
- Did the authors perform somatostatin-based imaging of the tumor?
- Was chromogranin A measured?
Author Response
Point-by-Point Responses to the Comments:
Dear Editors and Reviewers
Journal of Clinical Medicine
We wish to submit a revised our article for publication in Journal of Clinical Medicine, titled “A case of 123I-metaiodobenzylguanidine scintigraphy-negative pheochromocytoma with a tumor-developing mutation in the RET gene.” (Manuscript ID: jcm-1840781).
We have carefully considered the comments provided by the reviewers, and our point-by-point responses to each are included in response letter shown below. The respective changes made to the manuscript are written with “Track Changes” function as instructed with some minor corrections for language, grammar, and consistency. Revised manuscript is attached as Word file and please see the content. Thank you very much for reviewing our manuscript and offering valuable advice.
Reviewer #2:
The authors report a case of an MIBG negative pheochromocytoma where germ-line mutation analysis revealed a well-known polymorphism of the RET protooncogene. The clinical relevance of the Gly691Ser polymorphism is debated and it cannot be regarded as a pathogenic RET mutation. It is definitely not regarded as a causative mutation for the MEN2 syndrome.
Based on the inconclusive results on Gly691Ser - in my view - its pathogenic relevance cannot be established in this case either. The link between Gly691Ser and pheochromocytoma in this case is merely hypothetic and could also be an accidental association. It is unclear how the MIBG negativitiy of the PCC could be related to this polymorphism. The authors report no signs of MEN2. The presentation of the methods is inadequate.
Reply:
We appreciate for the important suggestion and advice. Previous reports using molecular biological procedure did not clarified the mechanism of MIBG negativity of pheochromocytoma and it is still unknown.
Biologically, MIBG-uptake was done through specific membrane-bound transporters, VMAT, which transports amines into storage vesicles. Thus, the decreased expression of VMAT results in MIBG-negative neuroendocrine tumors (L Kölby et al. Br J Cancer. 2003), although this is not confirmed in pheochromocytoma, probably due to disease rarity.
After further previous article research, we found that Lauren F et al. (Cell Tissue Res. 2018), Huynh, T.T. et al. (Eur J Endocrinol. 2005) and Annika M A Berends et al. (Cancers (Basel). 2019) reported VMAT expression is relatively low in RET mutated pheochromocytoma (kinase signaling group) than other types of pheochromocytoma (pseudohypoxia and Wnt signaling groups). Moreover, the deceased VMAT expression may suggest poorer cellular dedifferentiation, making certain tumors more aggressive.
Take these reports into considerations, the combination of the present RET mutation and pathologically moderately differentiated tumors accompanying pseudo rosette pattern, suggesting immature cells, may contribute to MIBG-negativity in this case.
Supplementary, we do agree the present RET mutation is apparent causative mutation for the MEN2 syndrome, but this mutation could alter kinase signaling from a viewpoint of enhancing signal transduction based on in silico analysis.
These discussions are added to the section of Discussion (lines 155-162, 166-168, 172-183, 189 and 196-198, page 6 and line 214-215, page 7)
Minor points:
- the normal ranges of hormones should be given
Reply:
Thank you very much for the suggestion. We newly added reference range of the laboratory data and edited Table 1 (page 3-4).
- the authors state in the introduction that 70 % of PCC are MIBG positive, but later in the discussion they mention 90-100 % sensitivity and specificity
We apologize for confusing sentences. The former sentences intended that “at least” 70% sensitivity or specificity is secured for pheochromocytoma in MIBG scintigraphy. Otherwise, latter parts intended that, in some reports, MIBG-scintigraphy showed up to 100% sensitivity and specificity. To clarify the meaning, we edited the introduction (line 45-46, page 2) and discussion section (line 154, page 6).
- Did the authors perform somatostatin-based imaging of the tumor?
- Was chromogranin A measured?
Reply:
We agree that these are important data. However, in Japan, somatostatin-based imaging for the gastrointestinal neuroendocrine tumor was solely covered by the Japanese health insurance and we do not have a chance to conduct somatostatin-based imaging for pheochromocytoma as research project. Therefore, we could not conduct somatostatin-based imaging in this case. Similarly, measuring chromogranin A is not covered by Japanese health insurance and we do not have a chance to measure chromogranin A.
We believe that these changes have greatly improved the quality of our paper, and we wish to thank you for your consideration. I look forward to hearing from you.
Sincerely,
Haremaru Kubo and Yuya Tsurutani
Endocrinology and Diabetes Center, Yokohama Rosai Hospital, Japan 3211 Kozukue-cho, Kouhoku-ku, Yokohama, Kanagawa, 222-0036, Japan.
Tel: +81-45-474-8111
Fax: +81-45-474-8323
e-mail: yuya97tsuru1055@gmail.com
Reviewer 3 Report
The case report by Kubo et al. presented a case of a 33-year-old man patient who was admitted to their hospital for further evaluation of hypertension. CT scan and biochemical tests’ results were suggestive of sporadic pheochromocytoma (PCC), even though MIBG scintigraphy was negative. Generally, MIBG scintigraphy-negative PCC exhibits a high frequency of SDHB mutation, however target gene testing in this patient revealed a mutation in RET gene, which is associated with potential for tumor malignancy. This suggested that RET gene mutations may be associated with MIBG scintigraphy-negative PCC when it does not also have express the SDHB mutation.
The article precisely described the clinical case and could be very interesting from a clinical point of view.
Only few minor concerns should be addressed by the Authors:
· Line 106: please explain briefly what is intended by “GAPP score”;
· Line 118: which type of genes are those?;
· “Material and Methods” paragraph: add the type of sequencing, the instrument, the kit and the protocol used in detail.
Overall. MINOR REVISIONS are required.
Author Response
Point-by-Point Responses to the Comments:
Dear Editors and Reviewers
Journal of Clinical Medicine
We wish to submit a revised our article for publication in Journal of Clinical Medicine, titled “A case of 123I-metaiodobenzylguanidine scintigraphy-negative pheochromocytoma with a tumor-developing mutation in the RET gene.” (Manuscript ID: jcm-1840781).
We have carefully considered the comments provided by the reviewers, and our point-by-point responses to each are included in response letter shown below. The respective changes made to the manuscript are written with “Track Changes” function as instructed with some minor corrections for language, grammar, and consistency. Revised manuscript is attached as Word file and please see the content. Thank you very much for reviewing our manuscript and offering valuable advice.
Reviewer #3:
The case report by Kubo et al. presented a case of a 33-year-old man patient who was admitted to their hospital for further evaluation of hypertension. CT scan and biochemical tests’ results were suggestive of sporadic pheochromocytoma (PCC), even though MIBG scintigraphy was negative. Generally, MIBG scintigraphy-negative PCC exhibits a high frequency of SDHB mutation, however target gene testing in this patient revealed a mutation in RET gene, which is associated with potential for tumor malignancy. This suggested that RET gene mutations may be associated with MIBG scintigraphy-negative PCC when it does not also have express the SDHB mutation.
The article precisely described the clinical case and could be very interesting from a clinical point of view.
Only few minor concerns should be addressed by the Authors:
Line 106: please explain briefly what is intended by “GAPP score”;
Reply:
Thank you for the important suggestion.
As Wachtel H et al. described in recent report (Wachtel H et al. J Clin Endocrinol Metab. 2020), GAPP score was established by Kimura N et al. and used worldwide for predicting metastatic potential in pheochromocytoma.
In the present case showed moderately differentiated GAPP score (4 points within range of 3-6 points). This score was significantly associated with disease recurrence and we must take more careful follow-up in this case with routinely check-up. These are added in the line 108-113, page 4.
Line 118: which type of genes are those?
We appreciate for the important advice. The molecular characterization of pheochromocytoma has been provided in some reports. There are three clusters are shown; 1) the pseudohypoxia group (including SDHB, VHL and HIF2A), 2) the Wnt signaling group (including CSDE1) and 3) the kinase signaling group (including RET, MAX and TMEM127). These are newly described in line 121-125, page 5.
In this case, the present RET gene mutation was found suggesting kinase signaling. The kinase signaling group mostly shows preservation of differentiated or matured cells, though this case showed moderately differentiation pathologically and this is interesting and important point in this case.
“Material and Methods” paragraph: add the type of sequencing, the instrument, the kit and the protocol used in detail.
Reply:
We additionally precisely described the sequencing methods in the section of “Material and Methods” (line 136-147, page 5).
About several genes (MAX, TMEM127, SDHB, CSDE1, and HIF2A), we conducted the procedure to the DNA amplification step in our laboratory. Amplification of DNA were conducted by an experienced technician. Thereafter, sequencing of purified DNA fragments was outsourced to Eurofins Genomics (Tokyo, Japan).
Otherwise, in Japan, RET and VHL genes sequencing were covered by the Japanese health insurance. Thus, we ordered the analysis through outsourcing service from the first step of gDNA extraction.
We believe that these changes have greatly improved the quality of our paper, and we wish to thank you for your consideration. I look forward to hearing from you.
Sincerely,
Haremaru Kubo and Yuya Tsurutani
Endocrinology and Diabetes Center, Yokohama Rosai Hospital, Japan 3211 Kozukue-cho, Kouhoku-ku, Yokohama, Kanagawa, 222-0036, Japan.
Tel: +81-45-474-8111
Fax: +81-45-474-8323
e-mail: yuya97tsuru1055@gmail.com
Round 2
Reviewer 2 Report
The manuscript has significantly improved, it is clearer. I still think that the association of this RET polymorphism with the current case can also be accidental, and I reckon that this should be included in the discussion. The authors should also state that it is a hypothesis they have presented.
